# Stretch Causes Cell Stress and the Downregulation of Nrf2 in Primary Amnion Cells

**DOI:** 10.3390/biom12060766

**Published:** 2022-05-31

**Authors:** Justin Gary Padron, Nainoa D. Norman Ing, Po’okela K. Ng, Claire E. Kendal-Wright

**Affiliations:** 1Anatomy, Biochemistry and Physiology, John A. Burns School of Medicine, University of Hawai‘i at Mānoa, Honolulu, HI 96822, USA; jpadron@hawaii.edu; 2Wayne State School of Medicine, Detroit, MI 48201, USA; 3Natural Science and Mathematics, Chaminade University of Honolulu, Honolulu, HI 96816, USA; nainoa.normaning@student.chaminade.edu (N.D.N.I.); spencerkamakane@gmail.com (P.K.N.); 4Obstetrics, Gynecology and Women’s Health, John A. Burns School of Medicine, University of Hawai‘i at Mānoa, Honolulu, HI 96826, USA

**Keywords:** amnion epithelial cell, Nrf2, stretch, NF-kB, HMGB1, sulforaphane, ROS

## Abstract

Nuclear-factor-E2-related factor 2 (Nrf2) is a key transcription factor for the regulation of cellular responses to cellular stress and inflammation, and its expression is significantly lower after spontaneous term labor in human fetal membranes. Pathological induction of inflammation can lead to adverse pregnancy outcomes such as pre-eclampsia, preterm labor, and fetal death. As stretch forces are known to act upon the fetal membranes in utero, we aimed to ascertain the effect of stretch on Nrf2 to increase our understanding of the role of this stimulus on cells of the amnion at term. Our results indicated a significant reduction in Nrf2 expression in stretched isolated human amnion epithelial cells (hAECs) that could be rescued with sulforaphane treatment. Downregulation of Nrf2 as a result of stretch was accompanied with activation of proinflammatory nuclear factor-kB (NF-kB) and increases in LDH activity, ROS, and HMGB1. This work supports stretch as a key modulator of cellular stress and inflammation in the fetal membranes. Our results showed that the modulation of the antioxidant response pathway in the fetal membranes through Nrf2 activation may be a viable approach to improve outcomes in pregnancy.

## 1. Introduction

The phenomenon of preterm birth (PTB) (birth <37 weeks’ gestation) remains the leading cause of perinatal morbidity and mortality, affecting 1 in 10 of all births worldwide [1]. In 2020, the rate of prematurity rose for the fifth year in the United States, posing risks for complications in surviving infants. This is significant because the sequelae of this may extend well into adulthood; resulting in cerebral palsy, retinopathy of prematurity, and cardiovascular disease [2,3]. Additionally, socioeconomic impacts of social isolation and financial burden have also been identified as negative outcomes of PTB [4]. The United States faces significant racial and ethnic disparities in PTB, whilst also having PTB rates much higher than other developed countries [1,5]. While PTB’s etiology is complex, infection leads to almost 50% of preterm deliveries [6]. Thus, infection along with other medically indicated causes of prematurity, such as pre-eclampsia and polyhydramnios caused by gestational diabetes, are associated with inflammation and oxidative stress [7,8]. However, we believe that understanding the normal mechanisms of parturition can provide insight into how these processes deviate in PTB. 

During normal pregnancy, the integrity of the fetal membranes, which is composed of the amnion and chorion that is adhered to the decidua, must be maintained until delivery [6]. Although the membranes often rupture during labor, this is preceded by biochemical and biophysical changes thought to weaken the membranes ahead of this event [6,9,10]. The biochemical changes that are seen to occur prior to membrane rupture include inflammation and oxidative stress, which are well-characterized for their roles in membrane weakening [11,12]. The intrauterine build-up of oxidative stress has been shown to accelerate the accumulation of inflammation leading to extracellular matrix component degradation and therefore fetal membrane weakening [12,13]. As an inducer of oxidative stress and inflammation, stretch forces are also important in the fetal membrane weakening process. The membranes are subject to pre-labor stretch forces by Braxton-Hicks contractions and have been shown to be increasingly distended in utero [14,15,16]. In support of this, fetal membrane overdistension implicated by multiple pregnancies and polyhydramnios is associated with an increased risk of PTB [17]. Thus, although the stretch of the amnion tissues is associated with an inflammatory pulse that precedes labor [18,19,20], the precise underlying mechanisms connecting them are clear.

Although much less studied in connection with parturition, nuclear factor erythroid 2 (NFE2)-related factor 2 (Nrf2) is a protein complex involved in the regulation of transcription of DNA, cell survival, and inflammation [21,22,23]. However, Nrf2 primarily plays a role in the anti-oxidative stress response induced by ROS generation [23,24,25]. Nrf2 has emerged as a regulator of oxidant resistance and has been implicated in toxicities and chronic conditions associated with oxidative stress [26]. Although Nrf2 is known to be activated by ROS and cell stress, severe stress conditions such as septic shock and high levels of oxidative stress paradoxically compromise Nrf2 expression resulting in increased inflammation and tissue injury [27,28,29]. There is increasing evidence to suggest that Nrf2 plays a major role in many pathways involving inflammation. In support of this, Nrf2 deficiency pathologies are associated with nuclear factor-kB (NF-kB)-mediated inflammation [30,31,32]. Nrf2 is a multifaceted as it has roles in many different disease states with robust inflammatory signatures such as cancer progression and chemoresistance [33,34]. However, Nrf2 also decreases with labor and its activation reduces cytokine IL-6 expression [35,36]. To the best of our knowledge, the influence of stretch on Nrf2 activity has not been explored specifically in the amnion cells of the fetal membranes. As high levels of oxidative stress and increased inflammation are associated with weakening of the fetal membranes, a role of Nrf2, specifically its downregulation, is implicated.

Therefore, in this study, we aimed to determine the effect of in vitro stretch on amnion epithelial cells’ stress and Nrf2. We hypothesized that the in vitro stretch would induce a cellular stress response that activates the proinflammatory mediators; ROS, HMGB1, and NF-kB, while downregulating Nrf2. Due to its role in cellular protection, elucidating the effect of in vitro stretch on Nrf2 can give insight into its role in the fetal membranes.

## 2. Materials and Methods

### 2.1. Tissue Collection and Culture of Human Amnion Epithelial Cells

Fetal membranes (total of 15) were obtained from term repeat cesarean section live births at Kapiolani Women and Children’s Hospital (Honolulu, HI, USA) with IRB approval. All tissues were examined by a pathologist for histological evidence of infection, and if positive, were excluded from further analysis. The human amnion epithelial cells (hAECs) were isolated as previously described [19,20]. In short, the amnion was peeled away from the chorion and subsequently washed with PBS. The tissue was then minced and incubated three times with 1g trypsin (SAFC; Buchs, Switzerland) in 500 mL of Dulbecco’s modified Eagle medium (DMEM) (Life Technologies Limited; Paisley, UK) at 130 rpm at 37 °C for 30 min. Resultant individual cells were pelleted by centrifugation, and they were suspended in media (F12:DMEM; 1:1 *v*/*v*) (Invitrogen; Carlsbad, CA, USA) with fetal bovine serum (FBS) (10%, *v*:*v*), and 1% penicillin (200 U/mL), and streptomycin 100µg/mL (Life Technologies; Burlington, ON, Canada).

### 2.2. Cell Culture and Treated Stretch Experiments

Cells were seeded at 1 × 10^6^ cells per well on a collagen IV-coated silicone lined stretch plate (Flexcell Inc., Burlington, NC, USA). After 4–10 days, the cells reached 70–80% confluency and the media was replaced with 0.5% FBS DMEM/F12 for 12–16 h. Fresh 0.5% FBS DMEM:F12 treated with 0, 1, and 2 μM sulforaphane (MilliporeSigma; Burlington, MA, USA) was added before addition of stretch stimulus. Cyclic stretch and release was performed using the Flexcell-FX5000TM tension system (Flexcell; Burlington, NC, USA). More details of the Flexcell experimental apparatus can be viewed (https://www.flexcellint.com/category/tension, accessed on 9 May 2022). The 20% cyclic stretch and release experiment was performed as previously described [19] for 4, 8, and 16 h with an interval of 27 s of 20% stretch, followed by 7 s of release of this stretch to 0%.

### 2.3. RNA Isolation and Quantitative Real-Time PCR

WISH cells (ATCC CCL25), human amnion mesenchymal cells (hAMCs), and hAECs were placed onto a 10 cm^2^ culture plate and grown to 80% confluency. Cell media was then replaced with 0.5% FBS DMEM/F12 for 12–16 h. Total RNA was isolated using the RNeasy Mini Kit (Qiagen; Valencia, CA, USA) in accordance with the manufacturer’s instructions. Placenta tissue was collected after removal of the fetal membranes from sites arranged in quadrants and flash frozen in liquid nitrogen. RNA isolation was performed on flash frozen samples after biopulverizing using the Cellcrusher (Cellcrusher; Cork, Ireland) and using the RNeasy Mini Kit. RNA quality and concentration were quantified using the NanoDrop^TM^ Lite Spectrophotometer (Thermo Fisher Scientific; Wilmington, DE, USA). Reverse transcription was performed to convert 0.5 μg of RNA to cDNA using the High-Capacity RNA-to-cDNA^TM^ Kit (Thermo Fisher Scientific Baltics UAB; Vilnius, Lithuania). Primers for *Nrf2* (Hs00975961_g1) and *GAPDH* (Hs02758991_g1) were purchased from Applied Biosystems TaqMan Assays on Demand (ABI; Foster City, CA, USA) and used in accordance with the manufacturer’s instructions. Each 96-well plate included a water blank and a reverse transcriptase blank. Real-time PCR was carried out on an Applied Biosystems StepOne Real-Time PCR System (ABI; Foster City, CA, USA). Each reaction was run in triplicate and the results were normalized to the expression of *GAPDH* in each sample.

### 2.4. Western Blotting

Extraction of nuclear protein was performed using the Nuclear Extract Kit (Active Motif; Carlsbad, CA, USA) in accordance with the manufacturer’s instructions. Whole cell lysate was harvested with a modified RIPA buffer (50 mM Tris, pH 7.4, 1% NP-40, 0.2% sodium deoxycholate, 150 mM NaCl, 1 mM EGTA, 1 mM sodium orthovanadate, 1 mM NaF, and Roche complete mini EDTA-free protease inhibitor cocktail (Roche; Indianapolis, IN, USA)). The Pierce BCA Protein Assay Kit (Thermo Scientific; Grand Island, NY, USA) was used to determine protein concentrations, as per manufacturer’s instructions. Samples were denatured at 95 °C for 5 min in 2× SDS Laemmli buffer with 5% ß-mercaptoethanol. Then, 15 and 20 micrograms of nuclear and whole cell protein lysate, respectively, were separated onto 8% polyacrylamide gels and transferred onto 0.45 mm nitrocellulose membranes (Thermo Fisher Scientific; Rockford, IL, USA). Membranes were incubated with a blocking buffer (5% skimmed milk or 2% gelatin in 0.1% PBS-T) overnight at 4 °C with slight agitation. Subsequently, membranes were incubated overnight at 4 °C with slight agitation using 1:500 Rabbit Polyclonal Anti-Nrf2, (ab137550; Abcam; Cambridge, MA, USA), 1:500 Rabbit Polyclonal p65 (06-418; EMD Millipore Corporation; Burlington, MA, USA), 1:1000 Rabbit Polyconal Anti-lamin-b1 (ab16048; Abcam Inc.; Cambridge, MA, USA), and 1:200,000 Anti-beta Actin antibody (ab8226; Abcam Inc.; Cambridge, MA, USA) diluted in a blocking buffer. After washing with PBS-T, membranes were incubated for 1 h with 1:3000 Goat Anti-Rabbit IgG (H+L) HRP conjugate antibody (Bio-Rad; Hercules, CA, USA). Blotted membranes were washed with PBS-T and developed with enhanced chemiluminesence (Amersham; Piscataway, NJ, USA) and exposed to hyperfilm (Amersham; Piscataway, NJ, USA). Developed films were analyzed via densitometry using ImageJ software (National Institute of Health; Bethesda, MD, USA). Nuclear fraction Nrf2 bands were normalized to lamin-b1 loading controls. Whole cell lysate Nrf2 bands were normalized to beta-actin loading controls.

### 2.5. Immunohistochemical Localization of NRF2

IHC was performed on paraffin sections (5 mm) using the Vectastain ABC kit (Vector Laboratories; Burlingham, CA, USA) as previously described [37]. In short, fetal membrane tissue sections were deparaffinized and hydrated through a series of decreasing alcohol (95%, 80%, 70%, and 60% EtOH). This was followed by antigen unmasking in a steamer (>80 °C, 10 mM citric acid buffer, pH 6.0) for 20 min. Endogenous peroxidase activity was blocked using 0.3% hydrogen peroxide in methanol for 30 min. This was followed by 1:100 Rabbit Polyclonal Anti-Nrf2 (ab137550; Abcam; Cambridge, MA, USA) incubation in 1% bovine serum albumin (EMD Chemicals Inc.; Gibbstown, NJ, USA) in PBS for 1 h at RT. Adjacent tissue served as 1:100 IgG control (R&D Systems; Minneapolis, MN, USA). NRF2 and IgG stained tissues were incubated with a biotinylated secondary antibody for 30 min, rinsed with PBS, and treated with the avidin-biotin-peroxide complex (ABC) (Vector Laboratories, Burlingame, CA, USA) for an additional 30 min. Tissue sections were counterstained with hematoxylin, and re-dehydrated. Slides were mounted with Permount (Fisher Chemicals; Pittsburg, PA, USA). Images were taken using the Nikon C1 Plus Ti Eclipse imaging system (Nikon Instruments; Melville, NY, USA).

### 2.6. Immunocytochemical Localization of Nrf2 and NF-kB Subunit p65

Primary AEC were seeded into 4-well chamber slides at 75,000 cells per well. The cells were grown to 60–80% confluency before fixation with 4% paraformaldehyde (J.T. Baker Inc.; Phillipsburg, NJ, USA) in PBS for 10 min. After washing with PBS, cells were permeabilized with 0.1% Trition X-100 (VWR International Inc., Radnor, PA, USA) in D-PBS (containing calcium-magnesium) for 15 min. Cells were blocked with 5% BSA in PBS for 1 h and subsequently incubated with 1:100 Rabbit Polyclonal Anti-Nrf2 (ab137550; Abcam; Cambridge, MA, USA) or p65 1:500 Rabbit Polyclonal p65 (06-418; EMD Millipore Corporation; Burlington, MA, USA) in 1% BSA in PBS for 1 h at RT. Cells were washed 3× for 5 min each with PBS and incubated with Alexa Fluor 488 anti-rabbit secondary (1:2000) for 1 h at RT. The cells were then washed with PBS and counterstained with DAPI (Molecular Probes, Inc.; Eugene, OR, USA) for 5 min before imaging with epi-fluorescence and confocal microscopy (Nikon Eclipse Ti, Melville, NY, USA). In addition, immunofluorescence was also performed on hAECs subjected to 4 h stretch and no stretch using the same antibodies. After counterstaining with DAPI for 1 min (1:50,000) and washing with PBS, the Flexcell membranes were cut around the edges with a scalpel and mounted in Mowiol/glycerol mounting media containing 1,4-diazabicyclo [2.2.2] octane (DABCO) (MilliporeSigma; Burlington, MA, USA) and viewed using the Nikon C1 Plus Ti Eclipse imaging system.

### 2.7. Lactate Dehydrogenase (LDH) Assay

Conditioned media from stretched and sulforaphane treated cells were added to 96-well plates and analyzed as per manufacturer’s instructions using The Pierce LDH Cytotoxicity Assay (Abcam; Cambridge, MA, USA). LDH activity was measured as the production of NADH, resulting in the change of absorbance at 450 nm using xMark Microplate Spectrophotometer (Bio-Rad; Hercules, CA, USA). LDH activity was measured every 15 s over the course of 30 min in units of mU/mL of LDH.

### 2.8. ROS Assay

Cytoplasmic fractions of treated and control samples collected using the Nuclear Extract Kit were subjected to ROS quantification using the Oxiselect^TM^ Intracellular ROS Assay Kit (Cell Bio Labs Inc.; San Diego, CA, USA) as per manufacturer’s instructions. ROS content was determined via dichlorofluorescin production in comparison with the predetermined DCF standard curve.

### 2.9. HMGB1 ELISA Assay

Conditioned media from stretched and sulforaphane treated cells were analyzed for HMGB1 secretion as per manufacturer’s instructions using the Human HMGB1 ELISA Kit (LifeSpan BioSciences Inc.; Seattle, WA, USA).

### 2.10. Statistical Analysis

Statistical analysis was carried out using the GraphPad Prism 8 software (GraphPad Software Inc.; San Diego, CA, USA). The paired Student’s *t*-test or ANOVA was performed with *p* values < 0.05 considered significant.

## 3. Results

### 3.1. Stretch Induces a Cellular Stress Response

As we have previously shown that the stretching of human amnion epithelial cells (hAECs) can increase the production of proinflammatory cytokines [19,20], we wanted to further characterize the cellular response by hAECs precipitated by this stimulus. The rate of lactate dehydrogenase (LDH) activity, a non-specific marker for cell stress, was first measured by the rate of NAD reduction to NADH. The results showed that 4 h of stretch significantly increased LDH activity by 39.72% (*n* = 5, *p* < 0.05) (Figure 1A). Because the LDH data provided evidence of cell stress by stretch, we then investigated the direct effect of stretch on the danger-associated molecular pattern, HMGB1. HMGB1 secretion was significantly increased by 18.1% (*n* = 4) in conditioned media with 4 h of hAEC stretch (Figure 1B). As the hAECs showed that the cells became stressed after stretching, the effect of it on the proinflammatory NF-kB cascade was measured by the measurement of NF-kB p65 subunit translocation to the nucleus with immunofluorescence. We found that 4 h of hAEC stretch significantly increased NF-kB p65 subunit nuclear translocation by 85% (*n* = 4, *p* < 0.05) (Figure 1C).

### 3.2. Nrf2 Is Expressed in Term in Human Fetal Membrane Cells and Its Expression Is Maintained after hAEC Isolation and Cell Culture

Nrf2 levels have been shown to decrease with labor in the fetal membranes [38]. Therefore, in order to then investigate the effect of stretch on Nrf2 expression using our model system, we sought to determine if the basal levels of Nrf2 expression at the term collection of our human fetal membranes were high enough for us to study its regulation. Qualitative immunohistochemical analysis of collected fetal membrane tissues (*n* = 3) showed robust Nrf2 expression throughout the cells of the amnion, chorion, and decidua (Figure 2A) in comparison to IgG (control) staining (Figure 2B). In the amnion, amnion mesenchymal cells (hAECs) qualitatively displayed the strongest Nrf2 expression signal relative to control (Figure 2C,D). However, both hAEC and hAMC Nrf2 signals had robust nuclear expression. We then confirmed Nrf2 expression at the gene level in the cells of the amnion at collection and compared it with Nrf2 expression in HeLa-contaminated amniotic epithelial-like cells (WISH), placenta tissue, and hAMCs (*n* = 3) (Figure 2E). WISH cells (*n* = 3) were chosen as a control cell line, known for their robust basal expression of Nrf2. We also measured the expression of hAMCs as these cells were seen to have strong expression by immunohistochemistry (Figure 2A). Both hAECs and hAMCs (CT: 25.6 ± 0.13 and 27.3 ± 3.4, respectively; mean ± SEM) showed lower gene expression levels than the WISH cell line (CT: 23.3 ± 1.3) but higher levels than isolated placenta tissue (Ct: 29.6 ± 0.11). hAECs had higher Nrf2 expression than hAMCs. This result mirrored that obtained via immunohistochemical analysis. Immunocytochemical analysis for fluorescein isothiocyanate (FITC)-labeled Nrf2 further confirmed expression in hAECs after cellular isolation (Figure 2F–H).

### 3.3. In Vitro Stretch Downregulates the Expression of Nrf2 in Human Amnion Epithelial Cells

As we confirmed a robust baseline expression of Nrf2 at the tissue and cellular level in hAECs (Figure 2D), we used this amnion cell type to determine the effect of mechanical stretch on this transcription factor. hAECs were initially subjected to 20% cyclic stretch for 4, 8, and 16 h. After 4 h of stretch, cellular levels of Nrf2 were significantly decreased by 57.0% (*n* = 5, *p* < 0.01), in comparison to the control of not stretched cells (Figure 3A). After 8 and 16 h of stretch, Nrf2 was decreased but there were no significant differences in Nrf2 protein expression because we also saw that in the not stretched control condition, Nrf2 expression had decreased. As 4 h of 20% cyclic stretch caused a significant decrease in Nrf2 expression in comparison to the control condition, the effect of the translocation of Nrf2 into the nucleus was studied. Similar to the results obtained by the Western blotting of whole cell protein levels, nuclear Nrf2 expression was significantly decreased by 39.5% with mechanical stretch (*n* = 4, *p* < 0.05) (Figure 3B). These results confirmed a decrease in Nrf2 levels as a result of mechanical stretching. As Nrf2 was downregulated after stretch and ROS production can downregulate Nrf2 activity, ROS production was therefore measured after stretch. The decrease in Nrf2 expression was accompanied by a 41.2% increase in ROS production after stretching (Figure 3C).

### 3.4. Sulforaphane Rescues Nrf2 Independent of ROS in Stretched Amnion Cells

Sulforaphane has been shown to attenuate oxidative stress and activate the cellular antioxidant defense by inducing the Nrf2 pathway. As Nrf2 has been implicated to play a role in proinflammatory pathways associated with preterm birth [39], 1 and 2 µM sulforaphane-treated cells were subject to mechanical stretch to investigate its effects on Nrf2 translocation and activity. First, the effect of sulforaphane treatment on cellular toxicity was quantified using LDH activity. There was no significant difference in LDH activity in the stretch or control conditions with 0, 1, or 2 μM sulforaphane treatment (*n* = 6, ns) (Figure 4A). Nuclear protein was then isolated to quantify nuclear translocation of Nrf2 by Western blotting. We stretched the hAECs for 4 h as it has previously been demonstrated that dimethyl maleate (DEM) is able to rescue the decrease in Nrf2 in this time frame. Therefore, after 4h of stretch of hAECs, 1 µM sulforaphane treatment significantly increased nuclear Nrf2 levels by 57.8%, whereas 2 µM sulforaphane treatment led to a significant increase in nuclear Nrf2 by 122.2% (*n* = 4, *p* < 0.05) (Figure 4B). As sulforaphane-ameliorated stretch induced a downregulation of Nrf2, its effect on ROS production was also investigated. Sulforaphane treatment had no significant effect on ROS production in hAECs, as ROS was significantly increased with stretch at all sulforaphane concentrations (*n* = 4, *p* < 0.05) (Figure 4C).

## 4. Discussion

This study builds upon our work that is focused on understanding the role of stretch on the fetal membranes at the end of gestation [19,20]. It is the first to ascertain the effect of mechanical stretch on the expression of the transcription factor Nrf2 and changes in the levels of HMGB1 secretion and intracellular ROS, specifically in hAECs (Figure 5). Our previous work showed that stretching leads to the increase in IL-6 and PBEF secretion from hAECs, and therefore combined with our new data, supports the general thesis that this form of cellular stress stimulus can be proinflammatory in the amnion [13]. In further support of our results, it has previously been shown that silencing of Nrf2 in primary amnion cells increases the expression and secretion of the pro-inflammatory cytokine IL-6 and the chemokine IL-8 [38]. Thus, the rescue in the decrease in Nrf2 by sulforaphane in our study demonstrates that Nrf2 activation can be manipulated to produce anti-inflammatory and antioxidant processes that are important for timely parturition and healthy pregnancy outcomes.

Stretch forces and oxidative stress are thought to induce a proinflammatory state that contributes to fetal membrane weakening. This can be explained by the ensuing infiltration of matrix metalloproteinase (MMP)-producing leukocytes that degrade the collagen that comprise the fetal membranes [6,35,38,40,41]. Our increased LDH activity data showed that our model for stretch causes cellular stress and potentially inflammation. This was confirmed by the increase also seen in HMGB1 secretion. This nuclear protein is secreted when cells detect a ‘danger’ signal, where it then interacts with receptors such as TLR4 to initiate inflammatory cascades [41,42,43,44]. HMGB1 has also been shown to increase in secretion from hAECs after exposure to other cell stressors such as cigarette smoke extract [44]. Thus, it is thought that this danger-associated molecular protein (DAMP), HMGB1, can be regularly measured in pregnancy as an indicator of the stress currently in the tissues of pregnancy. HMGB1 is a marker for intra-amniotic inflammation in women with preterm birth [45].

We selected the maximum percentage of stretch possible that the Flexcell system can generate, as the fetal membranes adhered to the underlying decidualized myometrium experience at as much as 70% stretch in vivo at term [16]. Thus, this does not accurately represent the extreme levels of physiological cell stress at the end of pregnancy and our data should be viewed through this lens.

In concordance with increased LDH activity and HMGB1 secretion, Nrf2 expression was significantly decreased with stretch. Nrf2 is known to modulate inflammatory signaling pathways associated with abnormal pregnancy, including pre-eclampsia [46]. However, for us to determine the effects of stretch on Nrf2 expression, we first needed to confirm hAECs to have quantifiable expression of Nrf2 in our experimental tissue, as its levels are known to decrease with high levels of cell stress such as labor (Figure 5C). As it is understood that cell stress increases due to hypoxia and inflammation in the third trimester of pregnancy, we confirmed that Nrf2 was still expressed at sufficient levels after fresh fetal membrane collection at scheduled cesarean section and then its continued high expression following subsequent cell isolation (Figure 2). This is important as stress conditions greatly vary the levels of Nrf2 (Figure 5) to unquantifiable levels, as seen in a previous study in the amnion [38].

ROS production can induce NF-kB and LDH activity; however, it is also well-established that ROS is a powerful inhibitor of Nrf2 activity [47]. Thus, the ability of stretch to increase the generation of ROS was determined in the hAECs. ROS was significantly increased with stretch in hAECs (Figure 4C), and these results support the idea that stretch can activate inflammation and downregulate antioxidant defense pathways observed at the end of pregnancy, as ROS production is also seen in the placenta with increased gestational age [16,48,49,50,51].

The activation of the Nrf2 pathway via polyphenols has been shown to ameliorate insults related to pregnancy disorders associated with proinflammatory NF-kB translocation through the establishment of a strong antioxidant intrauterine environment [52]. Sulforaphane is a phytochemical that is a known activator of Nrf2. Sulforaphane is thought to induce Nrf2 by inhibiting Kelch-like ECH associated protein 1 (KEAP1), a cytoplasmic protein that targets Nrf2 for ubiquitination and proteasomal degradation under basal conditions (Figure 5A) [53]. Nuclear Nrf2 expression was shown to be significantly increased with sulforaphane treatment. These results showed that sulforaphane prevents the inhibitory effects of stretch on Nrf2 activity and induces its nuclear translocation. However, sulforaphane treatment did not have a significant effect on ROS generation in the stretch or control conditions. This may be because the time interval may be insufficient to capture the downstream effects of ROS scavenging known to be induced after Nrf2 activation. In addition, there were no significant increases in cellular stress as measured by LDH activity with sulforaphane treatment, which may also indicate its potential as a safe therapeutic treatment. Other natural compounds such as curcumin and resveratrol have also been shown to decrease oxidative stress through Nrf2 activation [54,55].

Our stretch model also clearly demonstrated a significant induction of subunit NF-kB-p65 translocation to the hAEC nucleus, indicating that in vitro stretch is a robust stimulus for proinflammatory NF-kB-p65 activation. This is supported by another study showing that NF-kB DNA binding increased after stretching of hAECs [56]. Under severe levels of oxidative stress, Nrf2 was downregulated and its cytoplasmic inhibitor Keap1 promoted NF-kB activation (Figure 5) [47]. In cardiomyocytes, increases in Nrf2 decrease the secretion of HMGB1 and the downstream signaling molecule for NF-kB, MyD88 [45]. In turn, while NF-kB activation has also been shown to inhibit Nrf2, we believe the level of high oxidation status in our model can be attributed to the downregulation of Nrf2 and stimulation of inflammation in mechanically stretched hAECs.

Oxidative stress and inflammation are known to contribute to fetal membrane weakening and Nrf2 is well-studied for its role in the antioxidant response and the mediation of inflammation. This is supported by its ability to upregulate phase II enzymes that scavenge ROS and inhibit the NF-kB pathway [57]. In the amnion, Nrf2 was shown to exert anti-inflammatory effects through the inhibition of cytokine production [38]. It was also shown that in patients with preterm premature rupture of the membranes, an increase in ROS and decrease in Nrf2 is seen that is replicated with Nrf2-interfering RNA experiments in hAECs [57]. Decreases in Nrf2 was also shown in a LPS-treated PTB mouse model [37], further implicating the antithetical roles of Nrf2 and NF-kB at the end of pregnancy. Therefore, future studies will be focused on increasing our understanding of the balance and interactions between these two key transcription factors at the end of pregnancy.

We showed that the mechanism of stretch induces cell stress in the form of oxidative stress and DAMP production, while activating proinflammatory transcription factor NF-kB and downregulating by Nrf2 (Figure 5); thus, perhaps its downstream antioxidant pathway. In addition, we showed that sulforaphane is a robust Nrf2 activator in hAECs, rescuing the effect of stretch on this transcription factor. Although this phytochemical has been widely studied for its suppressive effects on tumor growth, its anti-inflammatory effects demonstrating its ability to induce Nrf2 activation in the fetal membranes have not been established. Therefore, potent Nrf2 activators, especially those found in readily available foods, may provide therapeutic benefits for women susceptible to PPROM through their abilities to buffer cell stress and inhibit inflammation. As it has been shown that Nrf2 expression is decreased in the fetal membranes after labor, Nrf2 loss may also play a pivotal role in the regulation of inflammation and oxidative stress that is crucial to parturition mechanisms, including the weakening of the fetal membranes.

## Figures and Tables

**Figure 1 biomolecules-12-00766-f001:**
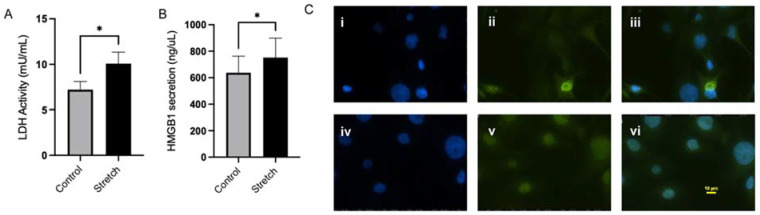
Mechanical stretch increased LDH activity, HMGB1 secretion, and activation of NF-kB. (**A**) Kinetically measured LDH activity (mU/μL) was increased with stretch. (**B**) ELISA quantification of HMGB1 (ng/μL) from cell culture supernatants of 4 h stretched and control of not stretched hAECs. Data displayed as mean ± STDEV. * *p* < 0.05 vs. control (Student’s *t*-test). (**C**) Immunofluorescent analysis of nuclear localization of p65 protein in hAECs. (**i**,**ii**) DAPI-blue (1µg/mL) and p65 (1/500) staining of control primary amnion epithelial cells, respectively. (**iv**,**v**) DAPI and p65 staining of stretched primary amnion epithelial cells, respectively. (**iii**,**vi**) Merged DAPI and p65 signal of control and stretched cells, respectively.

**Figure 2 biomolecules-12-00766-f002:**
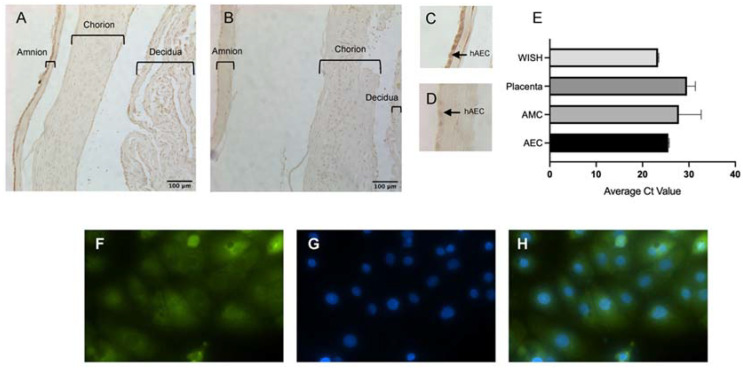
hAECs robustly expressed Nrf2 after human fetal membrane tissue collection and subsequent cell isolation and culture. (**A**) Representative immunohistochemical image of Nrf2 in fetal membranes, magnification 20× Nrf2 (**B**) IgG control. (**C**) Amnion layer of Nrf2 and (**D**) IgG control stained tissue amnion layer of IgG stained tissue. (**E**) qPCR analysis of Nrf2 expression normalized to GAPDH (*n* = 3). Data displayed as mean ± SEM. (**F**–**H**) Immunocytochemistry of Nrf2 expression in hAECs, magnification 60x FITC-labeled Nrf2, DAPI stain, and merged image, respectively.

**Figure 3 biomolecules-12-00766-f003:**
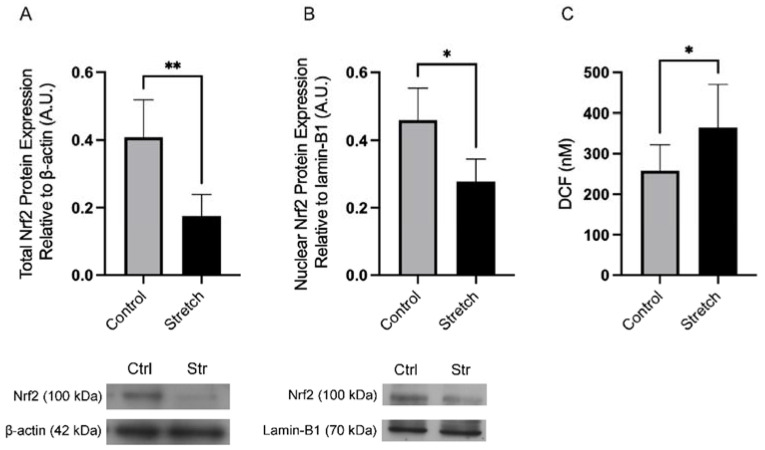
Stretch significantly decreased Nrf2 expression. (**A**) Whole cell lysate of Nrf2 (100 kDa) and β-Actin (42 kDa). (**B**) Nuclear lysate Western blot of Nrf2 (100 kDa) and Lamin-B1 (70 kDa) expression after 4 h of 20% cyclic stretch. (**C**) ROS generation as quantified by the production of DCF (nM) was increased with stretch. Data displayed as mean ± SEM. ** *p* < 0.01, * *p* < 0.05 vs. control (Student’s *t*-test).

**Figure 4 biomolecules-12-00766-f004:**
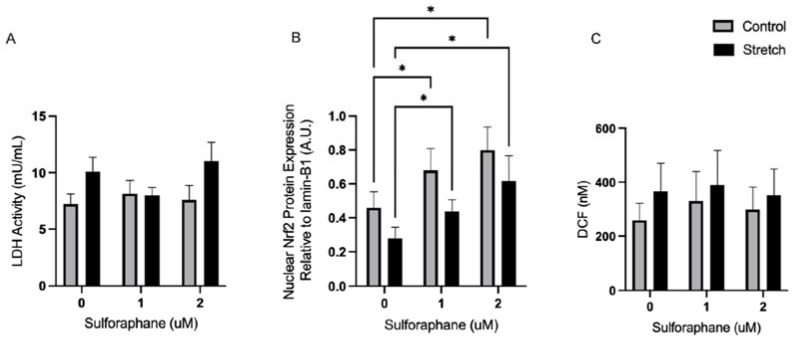
Sulforaphane rescued stretch-mediated Nrf2 downregulation independent of stretch. (**A**) Kinetically measured LDH activity (mU/μL) with sulforaphane treatment. (**B**) Nuclear Nrf2 protein expression relative to Lamin-B1 with sulforaphane treatment. (**C**) DCF production with sulforaphane treatment. Data displayed as mean ± STDEV. * *p* < 0.05 vs. control or untreated cells (two-way ANOVA).

**Figure 5 biomolecules-12-00766-f005:**
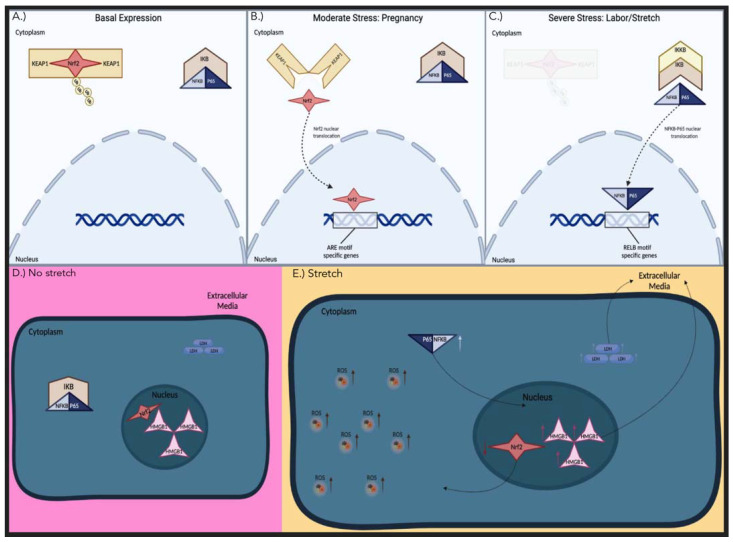
Summary of Nrf2 and NF-kB activity depending on the level of cell stress. (Top **A**–**C**) Activation of the transcription factors Nrf2 and NF-kB depending on the level of cellular stress. (**A**) Illustrates normal physiological conditions with no significant stress present. Ubiquitinated Nrf2 is bound to KEAP1, and NF-kB-p65 is bound to IkB, which inhibits their translocation into the nucleus. (**B**) The effects of moderate stressors such as the third trimester of pregnancy. Nrf2 dissociates from KEAP1, which allows for the nuclear translocation and binding to the ARE motif-specific genes. NF-kB-p65 remains bound to IkB in the cytoplasm. (**C**) Instances of severe stress equitable to labor and cell stretching. The elevated stress causes the eviction of the KEAP1-Nrf2 complex and activation of IKKB, which inhibits the inhibition of IkB, thus activating NF-kB-p65 and causing translocation of the dimer into the nucleus and binding to the RELB motif-specific genes. (Bottom **D**,**E**) Human amnion epithelial cells, (**D**) not stretched, (**E**) stretched 20%. Stretched cells had increased cytoplasmic ROS, increased secretion of HMGB1, and increased LDH activity. They also showed a decrease in total and nuclear Nrf2 and an increase in nuclear NF-kB-p65. Image created with Biorender (https://biorender.com, accessed on 9 May 2022).

## Data Availability

The data generated during the study are available upon request.

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
