# Peer review of "Stretch Causes Cell Stress and the Downregulation of Nrf2 in Primary Amnion Cells"

_biomolecules, 2022, doi:10.3390/biom12060766_

Round 1
Reviewer 1 Report
The work by Padron J, et al entitled "Stretch causes Cell Stress and the Downregulation of Nrf2 in Primary Amnion Cells" explores the effect of stretching forces on amniotic cells and its effect on the expression profile of Nrf2. The results showed that the stretch downregulates Nrf2, which was accompanied by activation of proinflammatory Nuclear factor-22 kB (NF-kB) and increases in LDH activity, ROS, and HMGB.
These results support the role of fetal membranes and specifically amniotic epithelium as a very sensitive and selective membrane capable of sensing its milieu and responding to change the immunological privilege in the maternal-fetal interface.
Introduction: This section is straightforward and offers an entire group of antecedents that permit understanding of mechanisms studied in this research. The lecture is easy, light, and enjoyable.
Material and Methods:
- In the material and methods section, a subsection of the "cell stretching experimental model" may be accompanied by a scheme showing the forces upon the cells attached to the culture system. To persons like me who are not familiar with this specific experimental procedure, it is complicated to "imagine" what is happening with the cells.
I consider that section 2.2 must be much more detailed. The current "abbreviate" format of this section makes it very difficult for the reader to understand the experimentation with the cells.
I had to search in a previous reference to understand the method. However, I'm not sure if the method used in the present work is similar to the reported in "Kendal-Wright, C.E.; Hubbard, D.; Gowin-Brown, J.; Bryant-Greenwood, G.D. Stretch and Inflammation-Induced Pre-B Cell 532 Colony-Enhancing Factor (PBEF/Visfatin) and Interleukin-8 in Amniotic Epithelial Cells☆. Placenta 2010, 31, 665–674, 533 oi:10.1016/j.placenta.2010.06.007.
I would like to see a scheme (maybe as a supplementary figure).
2. hAMCs should be defined in the manuscript. The author used human amniotic membranes-derived mesenchymal cells?.
I encourage the authors to explain the rationale for using hAMCs and WISH cells. What kind of information apport to their findings with the hAECs?
Results
- Figures 2A and 2B require scale and extra-explanation with arrows or brackets explaining the human fetal membranes' different regions and cell types. Photos 2A,2B, 2C, and 2D require improvement and look blurred.
- Figure 5, showing an integrative scheme with the different signal pathways involved in the stretch effect, is very appreciated. However, this scheme needs more design work; the letters are tiny, and it is very complicated to follow.
There are some typos along with the manuscript (IL-8 [34].. Thus)
Author Response
Reviewer 1:
The work by Padron J, et al entitled "Stretch causes Cell Stress and the Downregulation of Nrf2 in Primary Amnion Cells" explores the effect of stretching forces on amniotic cells and its effect on the expression profile of Nrf2. The results showed that the stretch downregulates Nrf2, which was accompanied by activation of proinflammatory Nuclear factor-22 kB (NF-kB) and increases in LDH activity, ROS, and HMGB.
These results support the role of fetal membranes and specifically amniotic epithelium as a very sensitive and selective membrane capable of sensing its milieu and responding to change the immunological privilege in the maternal-fetal interface.
Introduction: This section is straightforward and offers an entire group of antecedents that permit understanding of mechanisms studied in this research. The lecture is easy, light, and enjoyable.
Thank you for your kind encouragement.
Material and Methods:
- In the material and methods section, a subsection of the "cell stretching experimental model" may be accompanied by a scheme showing the forces upon the cells attached to the culture system. To persons like me who are not familiar with this specific experimental procedure, it is complicated to "imagine" what is happening with the cells.
I consider that section 2.2 must be much more detailed. The current "abbreviate" format of this section makes it very difficult for the reader to understand the experimentation with the cells.
I had to search in a previous reference to understand the method. However, I'm not sure if the method used in the present work is similar to the reported in "Kendal-Wright, C.E.; Hubbard, D.; Gowin-Brown, J.; Bryant-Greenwood, G.D. Stretch and Inflammation-Induced Pre-B Cell 532 Colony-Enhancing Factor (PBEF/Visfatin) and Interleukin-8 in Amniotic Epithelial Cells☆. Placenta 2010, 31, 665–674, 533 oi:10.1016/j.placenta.2010.06.007.
I would like to see a scheme (maybe as a supplementary figure).
This is indeed the same method used in this article from 2010. This equipment is used in many fields where epithelial cells are subject to distension forces. We have added more details into the method regarding the interval of stretching. We have also included a link to the company website in the methods that allows interested readers to view the equipment/cellular set up.
- hAMCs should be defined in the manuscript. The author used human amniotic membranes-derived mesenchymal cells?
This has been defined and yes, we collected fresh fetal membranes from repeat cesarean section, so the cells used in these experiments, described as hAMC or hAEC were primary cells.
I encourage the authors to explain the rationale for using hAMCs and WISH cells. What kind of information apport to their findings with the hAECs?
We used the WISH cells as a ‘control’ as we knew they had robust Nrf2 gene expression. We also measured the exprfessi8on of hAMC as they were shown to have strong expression levels by the immunohistochemistry. These cells are also important for the function and maintenance of the health of the amnion during pregnancy and so although the cell type was not the specific focus of this manuscript, we believed that this data would be interesting to our readers it the field.
Results
- Figures 2A and 2B require scale and extra-explanation with arrows or brackets explaining the human fetal membranes' different regions and cell types.
The figures have been amended to show the different areas.
- Photos 2A,2B, 2C, and 2D require improvement and look blurred.
These photos have been replaced and we hope you agree that they are much clearer now.
- Figure 5, showing an integrative scheme with the different signal pathways involved in the stretch effect, is very appreciated. However, this scheme needs more design work; the letters are tiny, and it is very complicated to follow.
Thank you, we are happy that the summary figure was useful, as we discuss many different molecules in the article. The text has been made larger and we hope you find this easier to follow now.
There are some typos along with the manuscript (IL-8 [34].. Thus)
Thank you, we have now reviewed the manuscript and remedied the typos.
Reviewer 2 Report
In this manuscript authors evaluated the effect of stretch on Nrf2 expression using term amnion cells. They found a significant reduction in Nrf2 expression in stretched hAECS that can be rescued with sulforaphane treatment. Downregulation of Nrf2 was accompanied with activation of proinflammatory NF-kB and increases in LDH activity, ROS, and HMGB1 proving that stretch is a key modulator of cellular stress and inflammation in the fetal membranes and Nrf2 plays a key role in this process.
Manuscript is clear and generally well written. However, some points need to be improved before publication. My comments are listed below.
- Introduction: Although authors properly introduced PTB, they did not mention important factors that can lead to PTB such as chorioamnionitis (PMID: 26739007) and polyhydramnios due to Gestational diabetes (PMID: 7651648). This is a very important point since both these conditions are associated to inflammation and oxidative stress.
- Line 59: Auhtors should stress the multifaceted role of NRF2 since it plays a key role in cancer progression and chemoresistance (see PMID: 35453348 and 34909662)
- Line 60: please remove 23-26 at apex
- Materials and Methods: the product code of primary antibodies must be reported
- Western blots: Authors must report the molecular weight in figures where Western Blot is shown
- Lines 389-391: Authors have to specify that, although these compounds have a key role in decreasing oxidative stress, only curcumin (PMID: 33477354) and Resveratrol (PMID: 28810692) have been reported to have a protective effects in preventing PTB. In fact, no beneficial effects were found regarding Vitamin C supplementation (PMID: 23508703). This point is very important since data reported by the authors in this paper could stimulate further studies regarding the use of these compounds in preventing PTB.
- An accurate revision of punctuation and typing errors is recommended
Author Response
Reviewer 2:
In this manuscript authors evaluated the effect of stretch on Nrf2 expression using term amnion cells. They found a significant reduction in Nrf2 expression in stretched hAECS that can be rescued with sulforaphane treatment. Downregulation of Nrf2 was accompanied with activation of proinflammatory NF-kB and increases in LDH activity, ROS, and HMGB1 proving that stretch is a key modulator of cellular stress and inflammation in the fetal membranes and Nrf2 plays a key role in this process.
Manuscript is clear and generally well written. However, some points need to be improved before publication. My comments are listed below.
- Introduction: Although authors properly introduced PTB, they did not mention important factors that can lead to PTB such as chorioamnionitis (PMID: 26739007) and polyhydramnios due to Gestational diabetes (PMID: 7651648). This is a very important point since both these conditions are associated to inflammation and oxidative stress.
Thank you for this suggestion. We agree that it really helps to illustrate the importance of understanding inflammation and oxidative stress in normal pregnancy and have therefore added text to the end of the first paragraph.
- Line 59: Authors’ should stress the multifaceted role of NRF2 since it plays a key role in cancer progression and chemoresistance (see PMID: 35453348 and 34909662)
We have added this to the introduction as we agree this helps to illustrate the broad range of disease states and literature in which this transcription factor is a key.
- Line 60: please remove 23-26 at apex
This has been removed.
- Materials and Methods: the product code of primary antibodies must be reported
Thank you for this suggestion, the catalog codes have now been added into the methods for all of the primary antibodies.
- Western blots: Authors must report the molecular weight in figures where Western Blot is shown.
We have added the molecule weight in kDa into figure 3.
- Lines 389-391: Authors have to specify that, although these compounds have a key role in decreasing oxidative stress, only curcumin (PMID: 33477354) and Resveratrol (PMID: 28810692) have been reported to have a protective effects in preventing PTB. In fact, no beneficial effects were found regarding Vitamin C supplementation (PMID: 23508703). This point is very important since data reported by the authors in this paper could stimulate further studies regarding the use of these compounds in preventing PTB.
We agree with the reviewers, as the data we have read about Vitamin C specifically in the Fetal membranes is contentious. Therefore in order to not mislead, we have removed vitamin C from this discussion.
- An accurate revision of punctuation and typing errors is recommended
Thank you for this prompt, we have now reviewed the manuscript for remaining errors